# Microstructure and Oxidation Behavior of Anti-Oxidation Coatings on Mo-Based Alloys through HAPC Process: A Review

**Tao Fu, Kunkun Cui, Yingyi Zhang *, Jie Wang, Xu Zhang, Fuqiang Shen, Laihao Yu and Haobo Mao**

School of Metallurgical Engineering, Anhui University of Technology, Maanshan 243002, China; ahgydxtaofu@163.com (T.F.); 15613581810@163.com (K.C.); wangjiemaster0101@outlook.com (J.W.); zx13013111171@163.com (X.Z.); sfq19556630201@126.com (F.S.); aa1120407@126.com (L.Y.); L1499923420@163.com (H.M.)
* Correspondence: zhangyingyi@cqu.edu.cn; Tel.: +86-173-7507-6451

**Abstract:** Mo and Mo-based alloys are important aerospace materials with excellent high temperature mechanical properties. However, their oxidation resistance is very poor at high temperature, and the formation of volatile $MoO_3$ will lead to catastrophic oxidation failure of molybdenum alloy components. Extensive research on the poor oxidation problem has indicated that the halide activated pack cementation (HAPC) technology is an ideal method to solve the problem. In this work, the microstructure, oxide growth mechanism, oxidation characteristics, and oxidation mechanism of the HAPC coatings were summarized and analyzed. In addition, the merits and demerits of HPAC techniques are critically examined and the future scope of research in the domain is outlined.

**Keywords:** molybdenum alloys; coating; oxidation; microstructure; mechanism; review



## 1. Introduction

Mo and its alloys have high melting point, excellent high-temperature mechanical properties, low thermal expansion coefficient and high conductivity and thermal conductivity, which have been widely used in high-temperature structural components in national defense industry, aerospace, and other fields, such as nozzle throat, high temperature electrode, high-temperature heating element, ray shielding material, etc. [1–4]. However, the oxidation resistance of Mo and Mo-based alloys is very poor, and they are easily oxidized to $MoO_3$ at a temperature of (400–800 °C) [5,6]. With the formation of $MoO_3$, the volume of molybdenum alloy increases rapidly, and leads to the occurrence of low temperature pulverization phenomenon, namely "Pesting oxidation". In addition, the formation of a large amount of volatile $MoO_3$ will lead to the catastrophic decomposition of molybdenum and its alloys when the oxidation temperature is greater than 1000 °C [7–10]. At present, alloying and surface coating can be used to improve the oxidation resistance of Mo and its alloys. Alloying is regarded as the preferred method to improve the properties of pure Mo, and Mo-based alloys have better mechanical properties than pure Mo when used at a high temperature above 1000 °C [11–14]. The classification, preparation method, properties, and application fields of molybdenum-based alloys are shown in Table 1 [15,16]. Because of limitations of alloying capability of Mo, its high temperature oxidation resistance cannot be fundamentally improved by alloying. Therefore, surface coating technology is regarded as an ideal method to improve the high-temperature oxidation resistance of molybdenum and its alloys [17–19]. Among them, the HAPC technology is the most widely used. At present, there are many reports about the oxidation behavior of HAPC coatings on Mo and its alloys. However, almost no reviewing of the progress in development of oxidation resistance of Mo has been documented [20].

**Table 1.** Classification and application of Mo-based alloys.

| Alloy Type | Preparation Method | Brand Number | Performance | Application Field | Refs. |
|---|---|---|---|---|---|
| Mo-Cu alloy | Co-deposition method; Metal oxide co-reduction, etc. | Mo-Cu | Good conductivity, thermal conductivity, ablation resistance, high hardness and strength | Electrician and electronics, instrumentation, national defense and military industry, aerospace, etc. | [15] |
| Mo-Ti-Zr alloy | Powder metallurgy; Smelting process | TZM, TZC | Excellent high temperature strength, high recrystallization temperature, good heat conduction and corrosion resistance | It is widely used in aerospace fields, such as rocket nozzles, nozzle throat liners, valve bodies, gas pipelines, etc. | [16] |
| Mo-Re alloy | Powder metallurgy; Vacuum smelting | Mo-5Re, Mo-41Re, Mo-50Re | Excellent radiation resistance and high tensile strength, good manufacturability and high temperature creep resistance | Aerospace, nuclear energy, chemical, electronics, military and so on. | [17] |
| Rare earth Mo alloy | Powder metallurgy | Mo-0.5Ti-Y, Mo-La, etc. | Good toughness, high temperature resistance, good bending resistance and tensile strength | High temperature furnace heating elements, nuclear materials, glass melting electrodes, etc. | [18] |

In this work, the advantages and disadvantages of the HAPC coatings are summarized and analyzed. The composition, exposure time, exposure temperature and mass change per unit area of the coatings have been given in relevant tables [21,22]. Their oxide growth mechanism and oxidation behavior are emphatically analyzed and summarized. Finally, the oxidation resistance and failure mechanism of the coatings are also summarized, aiming to provide some useful references for researchers in this field [23].

## 2. Microstructure and Growth Mechanism of HAPC Coatings

Halide activated pack cementation (HAPC) method is to embed the substrate into a mixture (Si powder, B powder, Al powder, $NH_4Cl/F$, Y, NaF, $Al_2O_3$ powder, etc.) with a certain particle size and proportion, and carry out thermal diffusion in vacuum or argon atmosphere to prepare diffusion coating. Figure 1 shows a schematic diagram of the HAPC reaction model. It can be seen that the $Al_2O_3$ crucible is usually used as the reaction device, and the plate on the top of the crucible plays the role of isolating air during the reaction process. The device is placed in a furnace and held at a set temperature for a certain period of time to obtain a coating on the substrate surface [24–29].

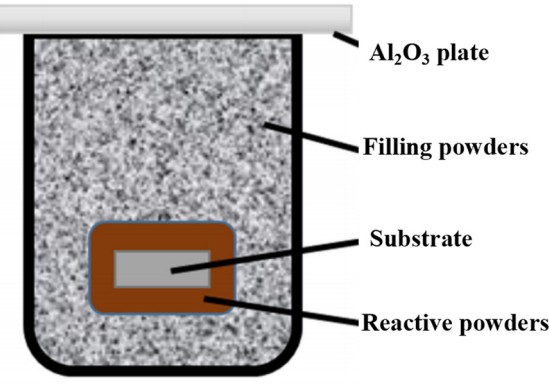

**Al$_2$O$_3$ plate**

**Filling powders**

**Substrate**

**Reactive powders**

**Figure 1.** Schematic diagram of the reaction model of the HAPC method. Reprinted with permission from [29]; Reproduced from (Yang et al., 2018).

Table 2 provides a summary of the process parameters, composition, and properties of HAPC coating on molybdenum-based alloys [29–42]. It can be seen that the composition of the mixture and the process conditions have an important influence on the microstructure, phase composition, grain size, and mechanical properties of the coating. Wang et al. [32] successfully prepared $MoSi_2$-MoB coatings with an average hardness of 5.84 GPa on Mo surface, and the hardness of MoB layer is as high as 9.54 GPa. By contrast, the surface hardness of pure $MoSi_2$ coating is only 2.58 GPa [34]. This is attributed to the addition of appropriate amount of B element in the mixed packing, which improves the fluidity of Si and makes the coating structure more dense and uniform [43]. The growth mechanism of HAPC coatings is shown in Figure 2. A $MoSi_2$ layer forms on molybdenum-based alloy, due to the interdiffusion reaction between Si powder and the substrate at high temperature. In the preparation process of coating, adding an appropriate amount of beneficial components (Al, B, YSZ, $ZrO_2$, $Al_2O_3$, SiC, MoB, etc.) can significantly improve the oxidation resistance and mechanical properties of the silicide coatings [44–49]. The main reaction equations involved in the above process are shown in Figure 2g.

**Table 2.** Summary of process, composition, and surface properties of HAPC coatings on molybdenum and its alloy.

| Substrate | Composition and Particle Size of HAPC Material | | Process Conditions | | Composition and Thickness (μm) | | Surface Hardness (GPa) | Grain Size of Coating Surface (μm) | Refs. |
|---|---|---|---|---|---|---|---|---|---|
| | Composition (wt.%) | Particle Size (μm) | Atmosphere | Treatment Time and Temperature | Outer Layer | Interface Layer | | | |
| Mo | C,Si,NaF | - | Air | 1200 °C, 2 h | $SiO_2$-$MoSi_2$ (55) | $Mo_5Si_3$ (5) | - | - | [29] |
| | 10Si-10$NH_4$F-80$Al_2O_3$ | 32.77 | Ar | 1300 °C, 10 h | $Al_2O_3$-$MoSi_2$ (60) | $Mo_5Si_3$ (1–2) | - | - | [30] |
| | 10Si-10$NH_4$F-80$SiO_2$ | 17.09 | Ar | 1300 °C, 10 h | SiO2-$MoSi_2$ (60) | $Mo_5Si_3$ (8–10) | - | 10–20 | |
| | 10Si-10$NH_4$F-80SiC | 4.87 | Ar | 1300 °C, 10 h | SiC-$MoSi_2$ (100) | - | - | 8–10 | |
| | 16Si-4B-4NaF–76$Al_2O_3$ | - | Ar | 1200 °C, 5 h | $MoSi_2$ (55–59) | $Mo_5Si_3$-MoB-$Mo_2$B (15–20) | - | - | [31] |
| | 20Si-0.8B-5NaF-74.2$Al_2O_3$ | - | Ar | 1000 °C, 10 h | $MoSi_2$ (27.2) | MoB (31) | 5.84 | - | [32] |
| | 16Si-4B-4NaF-2Y-76$Al_2O_3$ | - | Ar | 1300 °C, 5 h | $MoSi_2$ (190) | $Mo_5Si_3$-MoB-$Mo_2$B (14) | - | - | [33] |
| TZM | 7Si-87$Al_2O_3$-6$NH_4$Cl | - | Vacuum | 1000 °C, 12 h | $MoSi_2$ (100) | $Mo_5Si_3$ (2–3) | 2.58 | - | [34] |
| | 7Al-7Si-10$NH_4$F-76$Al_2O_3$ | ≤75 | Ar | 1100 °C, 17.5 h | $MoSi_2$ (100) | Mo(Si,Al)$_2$ (10) | - | - | [35] |
| | 12Si-3B-6Al-2$Y_2O_3$-5NaF-72$Al_2O_3$ | - | Ar | 1250 °C, 8 h | Mo(Si,Al)$_2$ (92) | MoB-$Mo_2$B (2–5) | - | - | [36] |
| | 7Al-7Si-10$NH_4$F-76$Al_2O_3$ | 1–2 | Ar | 1300 °C, 10 h | Mo(Si,Al)$_2$ (38) | $Mo_3$(Al,Si)-$Mo_5$(Si, Al)$_3$ (5–7) | - | 0.27 | [37] |
| | 70$Al_2O_3$-20Al-10$NH_4$Cl | - | Ar | 1000 °C, 12 h | $Al_2$($MoO_4$)$_3$ (20) | $Al_5$Mo-$Al_7$$Mo_4$ (30) | 2.58 | - | [38] |
| | 25Si-5NaF-70$Al_2O_3$ | - | Ar | 1100 °C, 6 h | $MoSi_2$ (35) | $Mo_5Si_3$ (2) | - | - | [39] |

**Table 2.** *Cont.*

| Substrate | Composition and Particle Size of HAPC Material | | Process Conditions | | Composition and Thickness (μm) | | Surface Hardness (GPa) | Grain Size of Coating Surface (μm) | Refs. |
| | Composition (wt.%) | Particle Size (μm) | Atmosphere | Treatment Time and Temperature | Outer Layer | Interface Layer | | | |
|---|---|---|---|---|---|---|---|---|---|
| | $81Al_2O_3$-$7Si$-$7Al$-$5NH_4F$ | ≤75 | Ar | 800–1000 °C, 8–36 h | $Mo(Si,Al)_2$ (20) | $MoSi_2$-$Mo_5Si_3$ (35) | - | - | [40] |
| Mo-30W | $7Al$-$7Si$-$5NH_4F$-$81Al_2O_3$ | ≤75 | Ar | 1000 °C, 16 h | Al-rich (12) | $(Mo,W)Si_2$ (46) | - | - | [41] |
| Mo-9Si-8B | $34.03Si$-$0.97B$-$2.5Na$-$62.5Al_2O_3$ | - | Ar | 1450 °C, 8 h | $MoSi_2$-$Mo_5Si_3$ (70) | $Mo_5SiB_2$-$MoB$ (10–15) | - | - | [42] |

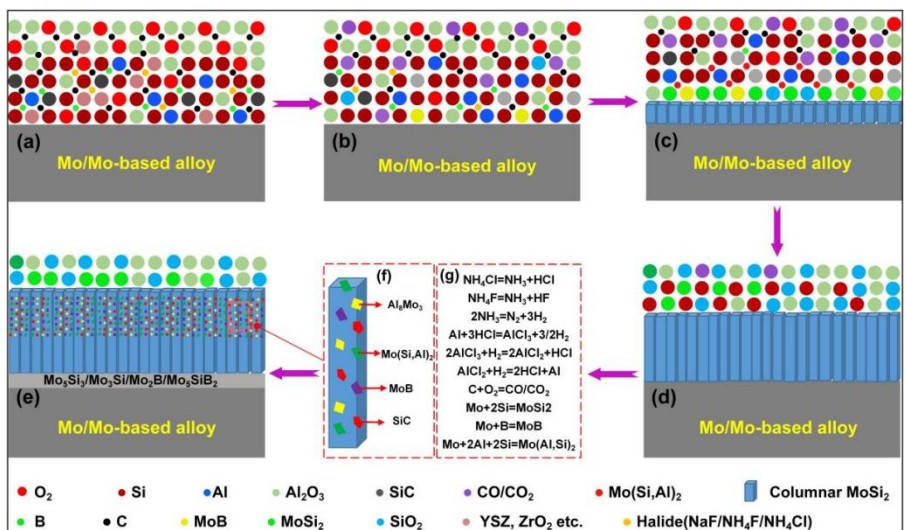

**Figure 2.** Growth mechanism diagram of HAPC coating on molybdenum and its alloys. (**a**) pre-sedimentation, (**b**) Oxygen consumption in the system, (**c**) The initial phase of the reaction, (**d**) The growth stage of the coating, (**e**) The final structure of the coating, (**f**) Amplification of individual coating grains, (**g**) The reactions involved in the above process.

The typical surface and cross-sectional morphology of the HAPC coatings are shown in Figure 3 [30]. It can be seen that some large particles are accumulated at the folds on the coating surface, which is mainly due to the large particle size of the mixture and uneven mixing, as shown in Figure 3a,c. The formation of micro-cracks is attributed to the thermal expansion mismatch between the coating and the substrate, as shown in Figure 3b,c. In addition, the addition of $Al_2O_3$, $SiO_2$, and SiC has great influence on the thickness of coating and interface layer. It is shown in Figure 3e that the $SiO_2$-$MoSi_2$ coating has a thick interface layer with a thickness of 8 to 10 μm. On the contrary, the thickness of interface layer of $Al_2O_3$-$MoSi_2$ coating is very thin and is only 1 to 2 μm. It is worth noting that the SiC-$MoSi_2$ coating has the thickest coating thickness, and the interface layer is not observed, as shown in Figure 3f. It can be seen that the addition of SiC significantly improves the deposition efficiency of the coating, which is mainly due to the physical deposition of SiC being faster than thermal diffusion deposition.

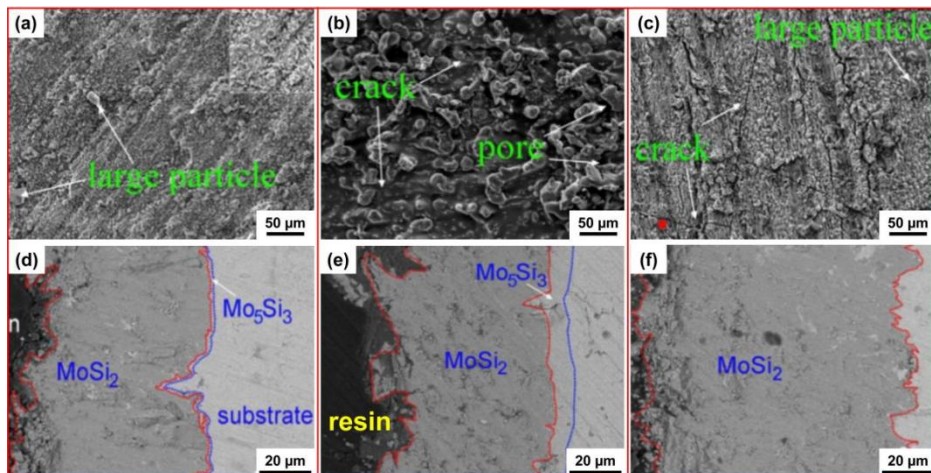

**Figure 3.** Surface and cross-sectional back scatter electronic (BSE) images of MoSi$_2$ coatings prepared with Al$_2$O$_3$ (**a**,**d**), SiO$_2$ (**b**,**e**), and SiC (**c**,**f**) filler in the pack, respectively. Reprinted with permission from [30]; Reproduced from (Sun et al., 2016).

## 3. Oxidation Behavior and Failure Mechanism of HAPC Coatings

The oxidation process parameters and oxidation properties of HAPC coatings on molybdenum and its alloy are shown in Table 3. It can be seen that the ingredients of the oxide layer mainly depend on the phase composition of the original coating. The oxide layer of pure MoSi$_2$ coatings is mainly composed of SiO$_2$ and Mo$_5$Si$_3$, however, composite coatings are mainly composed of SiO$_2$, Al$_2$O$_3$, and B$_2$O$_3$. The thickness of the outer coating decreases gradually with the increase of oxidation time. On the contrary, the thickness of the oxide layer and interface coating increases gradually, which is mainly due to the diffusion of oxygen and the internal diffusion reaction of silicon [29–42]. Figure 4 shows the typical surface and cross-section BSE images of oxidized coatings on TZM alloy. A large number of micro-cracks and MoO$_3$ particles are observed on the surface of the oxidized MoSi$_2$ coating, which is mainly due to the volume expansion of MoO$_3$, as shown in Figure 4a. However, the surface structure of Mo(Si,Al)$_2$ coating is complete without obvious cracks and pores after oxidation. This is due to the formation of Al$_2$O$_3$ with small thermal expansion coefficient during oxidation, which inhibits the pulverization of MoSi$_2$, as shown in Figure 5b [39]. The surface of the oxidized Mo(Si,Al)$_2$-MoB coating is very smooth and the oxide layer is clearly visible. This is attributed to the fact that SiO$_2$-Al$_2$O$_3$ generated in the oxidation process inhibits the volatilization of MoO$_3$, and MoB particles are dispersed and distributed inside the MoSi$_2$ coating, which makes the coating have a denser microstructure and better oxidation resistance, as shown in Figure 4c [38].

**Table 3.** Oxidation process parameters and oxidation properties of HAPC coatings on molybdenum and its alloys.

| Substrate | Composition and Thickness of Coatings (μm) | | Exposure | Composition and Thickness of Oxidized Coatings (μm) | | | Mass Gain (mg·cm$^{-2}$, wt.%) | Refs. |
|---|---|---|---|---|---|---|---|---|
| | Outer Layer | Interface Layer | | Oxide Layer | Intermediate Layer | Interface Layer | | |
| Mo | $SiO_2$-$MoSi_2$ (55) | $Mo_5Si_3$ (5) | 1600 °C, 1 h | $SiO_2$-$Mo_5Si_3$ | $MoSi_2$ | $Mo_5Si_3$ | 9.86 | [29] |
| | $Al_2O_3$-$MoSi_2$ (60) | $Mo_5Si_3$ (1–2) | 1200 °C, 110 h | $Al_2O_3$ (16) | $MoSi_2$ (56) | $Mo_5Si_3$ (42) | 0.15% | [30] |
| | $SiO_2$-$MoSi_2$ (60) | $Mo_5Si_3$ (8–10) | 1200 °C, 110 | $SiO_2$ (8–10) | $MoSi_2$ (32) | $Mo_5Si_3$ (45) | 0.05% | |
| | $SiC$-$MoSi_2$ (100) | - | 1200 °C, 110 h | $SiO_2$ (25–30) | $MoSi_2$ (64) | $Mo_5Si_3$ (40) | 0.28% | |
| | $MoSi_2$ (55–59) | $Mo_5Si_3$-$MoB$-$Mo_2B$ (15–20) | 1250 °C, 100 h | $SiO_2$-$B_2O_3$-$MoO_3$ (100) | $Mo_5SiB_2$ (20) | $Mo_5Si_3$-$MoB$-$Mo_2B$ (50) | 3.25 | [31] |
| | $MoSi_2$ (27.2) | $MoB$ (31) | 1300 °C, 80 h | $SiO_2$ (6-8) | $MoSi_2$-$Mo_5Si_3$ (50) | $MoB$-$Mo_2B$ (38) | 0.34 | [32] |
| | $MoSi_2$ (190) | $Mo_5Si_3$-$MoB$-$Mo_2B$ (14) | 1000 °C, 100 h, 1 h cycles | $SiO_2$ (16) | $MoSi_2$ (80) | $Mo_5Si_3$-$MoB$-$Mo_2B$ (32) | $1.33 \times 10^{-3}$ | [33] |
| TZM | $MoSi_2$ (100) | $Mo_5Si_3$ (2–3) | 1200 °C, 55 h | - | - | - | 0.15 | [34] |
| | $MoSi_2$ (100) | $Mo(Si, Al)_2$ (10) | 1100 °C, 250 h, 0.5 h cycles | $SiO_2$-$Al_2O_3$ (5–8) | $MoSi_2$ (45) | $Mo(Si,Al)_2$(20) | 0.08 | [35] |
| | $Mo(Si, Al)_2$ (92) | $MoB$-$Mo_2B$ (2–5) | 1400 °C, 25 h | $SiO_2$-$Al_2O_3$ (15–20) | $MoSi_2$-$MoB$ (88) | $Mo_5Si_3$-$Mo_2B$ (28) | 2.38 | [36] |
| | $Mo(Si, Al)_2$ (38) | $Mo_3(Al,Si)$-$Mo_5(Si, Al)_3$ (5–7) | 1100 °C, 10 h | $SiO_2$-$Al_2O_3$ (2.5) | $Mo(Si,Al)_2$ (50) | $Mo_3(Al,Si)$-$Mo_5(Si,Al)_3$ (20) | 12.92 | [37] |
| | $Al_2 (MoO_4)_3$ (20) | $Al_5Mo$-$Al_7Mo_4$ (30) | 1200 °C, 50 h | - | - | - | 0.15 | [38] |
| | $MoSi_2$ (35) | $Mo_5Si_3$ (2) | 1350 °C, 20 h | $SiO_2$ (1–2) | $MoSi_2$ (20) | $Mo_5Si_3$ (50) | 0.16 | [39] |
| | $Mo(Si, Al)_2$ (20) | $MoSi_2$-$Mo_5Si_3$ (35) | 1300 °C, 72 h | $SiO_2$-$Al_2O_3$ (5–10) | - | - | 0.694 | [40] |
| Mo-30W | Al-rich (12) | $(Mo,W)Si_2$ (46) | 1100 °C, 15 h | - | - | - | - | [41] |
| Mo-9Si-8B | $MoSi_2$-$Mo_5Si_3$ (70) | $Mo_5SiB_2$-$MoB$ (10–15) | 1300 °C, 100 h | $SiO_2$-$B_2O_3$ (25) | $MoSi_2$-$Mo$ (40) | $Mo_5Si_3$-$MoB$-$Mo_5SiB_2$ (75) | 3.82 | [42] |

In addition, Sun et al. compared the oxidation kinetics curves of the pure Mo and the $MoSi_2$ coatings prepared with $Al_2O_3$, $SiO_2$, and SiC filler in the pack, respectively, at different temperatures, as shown in Figure 5 [30]. It can be seen that after oxidizing at 500 °C for 110h, the quality of the deposited coating only changes from −0.18%–0.09%. Among them, the quality of the deposited SiC coating has little change before and after oxidation. However, the mass variation of Mo substrate under this condition is as high as 3%, as shown in Figure 5a,b. Furthermore, in the oxidation experiments at 1200 °C, the Mo substrate rapidly failed in the initial stage of oxidation, and the quality of the deposited $Al_2O_3$, $SiO_2$ coatings are Increases of varying degrees, respectively. However, the mass of deposited SiC coating increases first and then decreases under the oxidation conditions. This is caused by the formation and evaporation of oxidized carbon (CO, $CO_2$) during oxidation, as shown in Figure 5c.

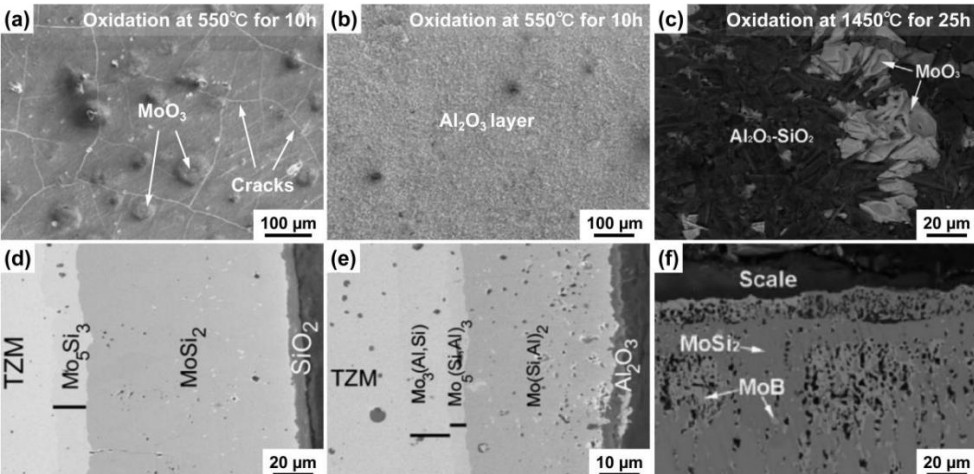

**Figure 4.** Surface and cross-section images of HAPC coatings at different Oxidation conditions; (**a**,**d**) $MoSi_2$ coating, (**b**,**e**) $Mo(Si, Al)_2$ coating. Reprinted with permission from [39]; Reproduced from (Paul et al., 2014). (**c**,**f**) $Mo(Si,Al)_2$-MoB coating. Reprinted with permission from [38]; Reproduced from (Chakraborty et al., 2008).

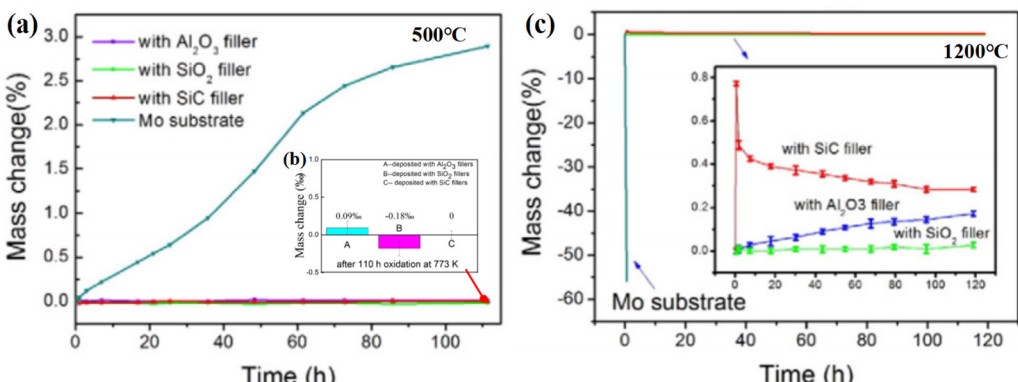

**Figure 5.** The oxidation kinetics in terms of time of exposure as well as temperature of Mo substrate and coatings at different temperatures, (**a**) 500 °C, (**b**) is the local amplification at 110 h as shown in (**a**,**c**) 1200 °C. Reprinted with permission from [30]; Reproduced from (Sun et al., 2016).

The microstructure evolution and oxidation mechanism of the coating are shown in Figure 6. Generally, the oxidation of the coating can be divided into transient oxidation and steady oxidation [50]. At the initial stage of oxidation, Mo, Si, B, Al, and other elements are oxidized at the same time. The quality loss of the coating is faster with the formation and volatilization of $MoO_3$, and the $SiO_2$ layer is discontinuous at this stage. At the same time, a large number of pores were observed on the coating surface after the evaporation of $MoO_3$. With the progress of oxidation reaction, continuous and dense oxide films ($Al_2O_3$, $SiO_2$, $B_2O_3$-$SiO_2$, etc.) gradually form on the coating surface. At last, a large number of pores are closed by $SiO_2$ with a low oxidation rate, and the oxidation process changes to a steady state [51,52].

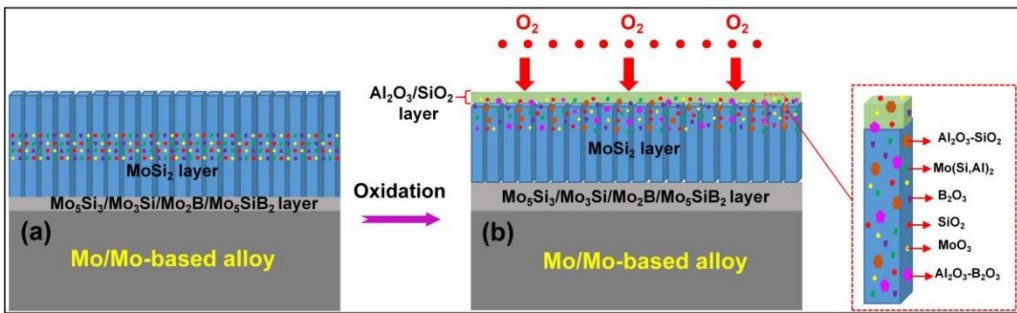

**Figure 6.** Oxidation mechanism diagram of HAPC coatings on molybdenum and its alloys. (**a**) Initial oxidation stage, (**b**) Late oxidation stage of coating.

## 4. Conclusions and Prospects

In this work, the growth mechanism, oxidation behavior, and mechanism of HAPC coatings are analyzed and discussed. The process conditions and properties of the coatings are provided. During the process, due to the relatively low process temperature of HAPC methods, the deposition efficiency of the coating is low, and the preparation time is long. However, the application of the processes is not limited by the shape of the substrate, and the coatings prepared have a uniform composition and good adhesion to the substrate. Figure 7 provides a summary of the composition and oxidation characteristics of protective coatings prepared by many researchers on the surface of molybdenum and its alloys by HAPC method.

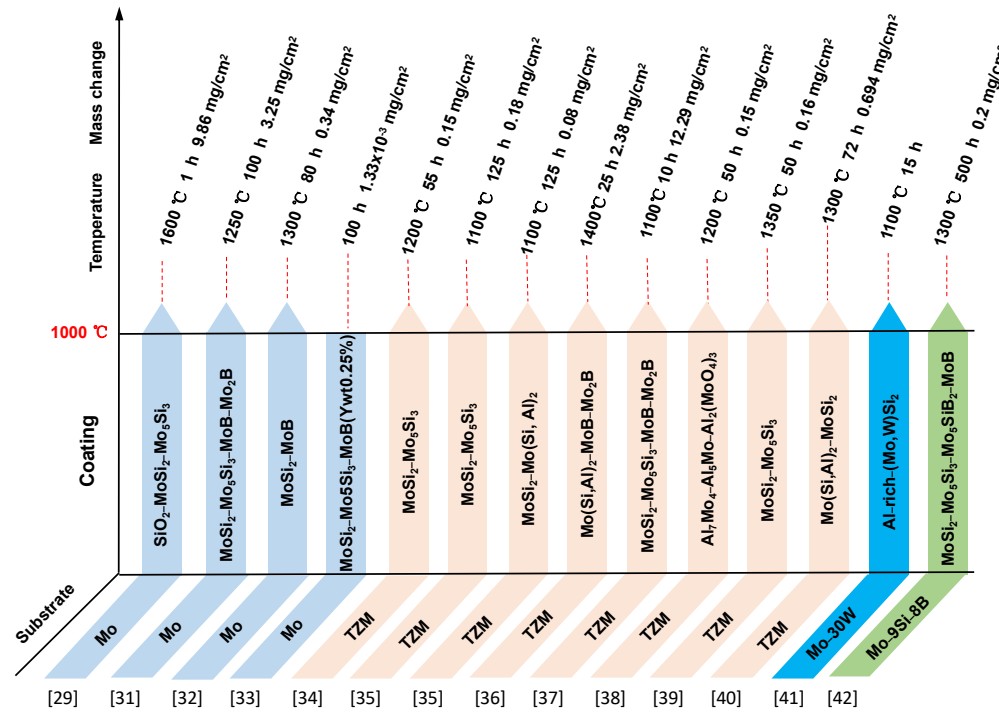

**Figure 7.** Overview of the composition and oxidation characteristics of HAPC coatings on the surface of Mo and its alloys.

In addition, the addition of beneficial elements and the second phases can not only significantly improve the mechanical properties and high temperature oxidation resistance of the coating, but also can delay the diffusion of silicon to the substrate, inhibit the formation of $Mo_5Si_3$ with poor oxidation resistance, reduce the formation of volatile $MoO_3$, and promote the rapid formation of continuous and dense anti-oxidation film on the

coating surface. The addition of Al element can significantly inhibit the formation of the pulverization $MoO_3$ and the occurrence of pulverization, which is mainly due to the formation of $Al_2O_3$ with low thermal expansion coefficient. The addition of B element can improve the oxidation resistance of the coating mainly in the following two aspects. On the one hand, in the process of high temperature oxidation (above 1400 °C), an appropriate amount of B dissolved into the oxide layer can further improve the fluidity of $SiO_2$, which promotes the healing of pores and cracks. On the other hand, MoB reacts with Si to produce $Mo_5SiB_2$ with low diffusion coefficient, which delays the diffusion of Si to the substrate and improves the oxidation life of the coating. In addition, Cr and W elements can also improve the oxidation resistance of the coating, which is mainly due to the formation of $Cr_2O_3$ with high oxidation resistance, and the formation of $(Mo,W) Si_2$ phase inhibits the internal diffusion of silicon into the substrate.

In order to improve the mechanical properties and antioxidant properties of the coating, the microstructure and phase composition of the coating must be optimized, and the addition of the second reinforcing phase also plays a positive role. Moreover, the multi-step coating preparation process can organically combine the single coating preparation technology with each other, overcome the limitations of its single application, and effectively extend the oxidation service life of the coating. This will be the direction of future research and development in this field.

**Author Contributions:** The manuscript was written through contributions of all authors. Y.Z.: Conceptualization, Investigation, and Supervision. Y.Z. and T.F.: Writing original draft and image processing. T.F., K.C., L.Y. and J.W.: Validation, Resources, Investigation, Writing-review & editing. X.Z., H.M. and F.S.: Visualization, Writing-review & editing. All authors have given approval to the final version of the manuscript.

**Funding:** This research was funded by the National Natural Science Foundation of China, Grant No. 51604049.

**Institutional Review Board Statement:** Not applicable.

**Informed Consent Statement:** Not applicable.

**Data Availability Statement:** The study did not report any data.

**Conflicts of Interest:** The authors declare that they have no known competing financial interests or personal relationships that could have appeared to influence the work reported in this paper.

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
