# Peer review of "Microstructure and Oxidation Behavior of Anti-Oxidation Coatings on Mo-Based Alloys through HAPC Process: A Review"

_coatings, doi:10.3390/coatings11080883_

Round 1

Reviewer 1 Report

  1. 'Growth mechanism' need to be replaced by oxide growth mechanism  in Abstract, conclusion, line 47 and else where. Only' growth mechanism' is misleading term
  2. Correct following English- in Abs.line 11 need to be reqritten as Extensive research on the poor oxidation problem has indicated that HPAC technology ……..
  3.  Line 15/16 in Abstract.. poor English and should be reworded as merits and demerits of HPAC techniques are critically examined and future scope of research in the domain is outlined.
  4.  Line 36, should be reworded as  Because of limitations of alloying capability of Mo, its high temperature ox. resistance....
  5. line 42, poor English  and needs rewriting as  Almost no reviewing of the progress in development of ox. resistance of Mo have been documented.
  6. Table 1  Reference is missing for each alloy type !!
  7. Fig 1 and 4   Reference is required  for each Figs. even if these were self constructed . Both Figs. deserve more elaborate discussions to highlight the mechanisms of coating development as this aspect is most important and key to understand the mechanisms. 
  8.  For readers to follow, the paper requires  short sections on the oxidation process and also for HPAC technology
  9. Fig. 2, Has the authors taken permission to reproduce from publishers/authors?? 
  10. Author should include a schematic illustration of different layers of coating formed on Mo/alloy and how the oxidation behavior is characterized, namely weight change/ thickness changes etc.
  11.  There is no plots showing the oxidation kinetics in terms of time of exposure as well as Temperature ? Most important for evaluation of the oxi. resistance.
  12. Line 99, what is meant of word' deserved' -makes no sense.
  13.  Table 3 shows just one reference for each of the alloy coating system--- more references should be consulted and included.
  14.  Oxidation behavior and mechanism of failure form the herats of the review work. Each one not dealt with properly and demands more elaborate discussions. Quantitative discussions should be included considering both cracks and pores e.g. size, density, distributions during oxidation. 

Reviewer 2 Report

The article is devoted to the analysis of data available in the open literature on the microstructure and chemical composition of layered coatings on molybdenum and molybdenum alloys, which reduce the oxidation rate at temperatures from 1100 ° C to 1600 ° C. The article is within the scope of the journal Coatings.

1.The problem of improving the functional and mechanical properties of high-temperature antioxidant coatings on molybdenum alloys remains urgent in practical terms. The results of the analysis of the available experimental data carried out by the authors of the article indicate the possibility of selecting such chemical compositions of coatings that are capable of providing an increase in antioxidant properties during high-temperature oxidation and preventing the formation of microcracks. The authors showed that one of the ways to increase the antioxidant properties of multilayer coatings with a changing phase composition is the formation of multiphase systems with Al2O3 and MoV particles inside the particles of the MoSi2 phase.

2. It should be noted the possibility of improving the quality of the article and its information content.

3. It is advisable to more clearly structure the article to substantiate the conclusions and conclusions formulated by the authors.

4. Apparently, the results of the analysis of information on the composition and characteristics of oxidation of the considered coatings on molybdenum and its alloys should be contained in the sections of the article, and not in the conclusion.

5. It is necessary to provide specific data for the conclusion of the authors on the improvement of the mechanical properties of antioxidant coatings on modibdenum and its alloys due to the formation of a second reinforcing phase.

6. This statement by the authors is not entirely specific and insufficiently supported by data analysis.

7. It is necessary to additionally substantiate the possibility of generalizing the data of the cited articles devoted to antioxidant coatings on other materials, including steels, tungsten and niobium alloys, with the results for modibdenum and its alloys [13, 14, 18, 19, 22, 23, 24 28, 46, 48, 50, 52],.

Reviewer 3 Report

The peer-reviewed article makes a good impression due to the systematic analysis of the existing technologies for the oxidation of molybdenum-based alloys. This allowed the authors to take into account the existing experience and clearly build their logic for the experiments. I would like to note the high quality and correct interpretation of metallography.

1. But the definition of properties is missing. For example, measuring the microhardness in depth of synthesized coatings, determining the adhesion of coatings to a substrate, etc.

2. I hope the authors will take these wishes into account for further research.

Author Response

Thank you very much for your comments.

Round 2

Reviewer 1 Report

Accept in present form